# Integrative Analysis of the Transcriptome and Metabolome Reveals the Developmental Mechanisms and Metabolite Biosynthesis of the Tuberous Roots of *Tetrastigma hemsleyanum*

**DOI:** 10.3390/molecules28062603

**Published:** 2023-03-13

**Authors:** Suni Hang, Pan Xu, Sheng Zhu, Min Ye, Cuiting Chen, Xiaojun Wu, Weiqing Liang, Jinbao Pu

**Affiliations:** 1School of Pharmaceutical Sciences, Zhejiang Chinese Medical University, Hangzhou 310053, China; 2Zhejiang Academy of Traditional Chinese Medicine, Hangzhou 310007, China; 3Zhejiang Guangsheng Pharmaceutical Co., Ltd., Quzhou 324022, China

**Keywords:** *Tetrastigma hemsleyanum* Diels et Gilg, transcriptome, metabolome, root development, metabolite biosynthesis

## Abstract

The tuberous root of *Tetrastigma hemsleyanum* Diels et Gilg (*T. hemsleyanum*) is a traditional Chinese medicine with a wide range of clinical applications. However, the scarcity of its wild resources, its low yield, and the variable quality that results from its artificial cultivation leads to expensive market prices that are not conducive to the further industrial development of *T. hemsleyanum*. In this study, transcriptomic and non-targeted metabolomic analyses were integrated to explore the underlying molecular mechanisms and metabolite biosynthesis that occur during its root development. Kyoto Encyclopedia of Genes and Genomes (KEGG) pathway enrichment analysis revealed that differentially expressed genes (DEGs) were predominantly enriched for processes associated with flavonoid and phenylpropanoid biosynthesis, starch and sucrose metabolism, and plant hormone signal transduction. Genes related to lignin were downregulated in tuberous roots (TRs), resulting in a decrease in lignification and the downregulation of metabolites related to flavonoids and phenylpropanoid biosynthesis. In addition, the expression levels of starch- and sucrose-related genes were upregulated in TRs. The root development of *SYQ* is also related to IAA, GA, ABA, and JA signaling pathways. Collectively, this study lays the foundation for analyzing the root development and quality-modulating mechanisms employed by *T. hemsleyanum*; this will be beneficial in conducting molecular-assisted breeding and controlling its secondary metabolite production.

## 1. Introduction

*Tetrastigma hemsleyanum* Diels et Gilg (Sanyeqing, SYQ) is an herbal medicinal plant that is endemic in China. The plant can grow for many years under natural conditions. During its growth period, there are only fibrous roots (FRs) in the early stage that then gradually expand to form tuberous roots (TRs). The TRs are usually regarded as the primary medicinal part of SYQ [1,2]. TRs have various pharmacological activities, including anti-inflammatory, antioxidant, antiviral, antitumor, antipyretic, anti-hepatic injury, immune regulation, and antibacterial activities, among others [3]. It can be used clinically to treat pneumonia, hepatitis, sepsis, children with high fever [4], and viral meningitis [5]. At present, wild resources of SYQ are scarce due to the strict requirements of their growing environment and over-excavation by human beings. Unfortunately, artificial cultivation requires a long growth period, often consisting of three years of planting before harvesting. Further problems of low yield and variable quality are also critical obstacles that hinder the production of the plant. Consequently, to improve the quality and yield of SYQ TRs, it is of significant importance to elucidate the developmental mechanisms and metabolite biosynthesis employed by TRs to form an essential theoretical basis for their improved production.

The initiation and development of TRs are complicated biological processes that are influenced by both internal and external factors. The TRs develop from adventitious roots. First, primary cambium cells between the primary phloem and the primary xylem of FRs form the circular primary cambium [6]. Subsequently, secondary meristems differentiate into the xylem. Simultaneously, the cells in the secondary cambium and meristems divide and swell; this is accompanied by the division of parenchyma cells in the xylem, resulting in root swelling. At this swelling stage, parenchyma cells accumulate a significant amount of starch. In order to facilitate the study of the initial and subsequent formation processes of root development, three types of root samples belonging to different developmental stages (consisting of non-swelling roots, initial swelling roots, and stable swelling roots) of SYQ were selected for analysis in this study.

The analysis of the development process of TRs is beneficial for improving their yield; improving the content of their active pharmaceutical ingredients is equally important. TRs have various chemical components, including flavonoids, terpenoids, steroids, polysaccharides, phenolic, organic, and fatty acids [7]. Numerous studies have shown that flavonoids comprise an important component of their pharmacological activity, and their synthetic pathways have been analyzed [8,9]. These flavonoids are synthesized from L-phenylalanine to produce *p*-coumaroyl-CoA through the common upstream steps catalyzed by phenylalanine ammonia-lyase (PAL), cinnamate 4-hydroxylase (C4H), and 4-coumarate-CoA ligase (4CL). Then, naringenin or liquiritigenin are biosynthesized from p-coumaroyl-CoA by chalcone synthase (CHS) and chalcone isomerase (CHI). Flavones, flavanones, isoflavones, anthocyanins, flavonols, and flavonoids are biosynthesized by branching [10].

In recent years, high-throughput methods have been widely applied to investigate the biosynthetic pathways associated with root development [11,12,13]. Transcriptomics can detect the expression changes of candidate genes during root development; this is then used to identify critical genes. In root development, metabolites, representing the terminal products of cell regulation, can arguably best reflect the phenotypic changes in biological systems. Therefore, the combination of transcriptomics and metabolomics can be used to provide precise information on interactions between structural genes and metabolites; this, in turn, facilitates the development of a high-resolution map of gene functions and metabolic pathways underlying certain biological processes. To date, research conducted on SYQ has primarily investigated pharmacological properties and involved cultivation techniques. However, there have been fewer studies on the molecular basis of gene expression levels and metabolite changes during the growth and development of SYQ TRs. Therefore, combined metabolomic and transcriptomic analysis can be conducted to establish the complete network of relationships between such genes and metabolites, thereby fully elucidating the developmental mechanisms of SYQ TRs.

In this study, we employed transcriptomics and non-targeted metabolomics to explore the transcriptional regulation and metabolic changes that occur during the maturation of SYQ roots. The flavonoid and phenylpropanoid biosynthesis pathways, starch and sucrose biosynthesis pathways, and plant hormone signal transduction pathways were explored using transcriptional metabolic association analysis. The results provide a basis for analyzing the mechanisms underlying SYQ TR development and the factors responsible for modulating their quality, further providing a scientific basis for their molecular breeding and chemical regulation.

## 2. Results

### 2.1. Morphological Changes at Different Developmental Stages of SYQ TRs

According to the root phenotypic characteristics exhibited by SYQ TRs at different developmental stages, it was found that the T1 samples were non-swelling roots, the T2 samples were initial swelling roots, and the T3 samples were stable swelling roots (Figure 1A). The microsection results showed that the number of cambium cells increased with the prolonged development of the roots. The vascular bundles of the xylem and phloem gradually degenerated over time, while the number of lignified cells around the xylem bundles also continued to decrease (Figure 1B). Meanwhile, the numbers and volumes of both the parenchyma cells and starch granules increased. According to the results of tissue micro-quantitative analysis (Table 1), it was found that the volumes of the vessels and parenchyma cells per unit of area increased. The starch granule numbers also increased when the roots became mature. The results of this experiment are consistent with the anatomical observations performed in this study, so we can infer that the continuous division of the cambium causes the fibrous roots to swell rapidly and that the appearance of the cambium cells is a signal of root formation.

### 2.2. The Content of Metabolites in Roots at Different Developmental Stages

The starch content increased significantly with continued root development (Figure 2A). The total flavonoid contents of the SYQ roots increased during the T2 stage and thereafter increased more slowly during the T3 stage (Figure 2B). Catechin, kaempferol-3-O-rutinoside, and astragalin were determined to be the primary flavonoids in the roots of SYQ. Its contents increased significantly in the T3 stage. The content of astragalin initially increased and subsequently declined.

### 2.3. Transcriptome Data Analysis

#### 2.3.1. Annotation and Expression of the Unigenes

A transcriptomic dataset of the SYQ roots at different developmental stages was constructed and the results were summarized. A total of 187,958 transcripts and 114,810 unigenes were obtained. All the unigene assemblies were annotated into six public databases, including nonredundant protein sequences (NR), Swiss-Prot, Pfam, Clusters of Orthologous Groups (COG), Gene Ontology (GO), and the Kyoto Encyclopedia of Genes and Genomes (KEGG). The most annotated database was NR, which annotated 59,355 unigenes, accounting for 51.83% of the total number of unigenes. The GO, KEGG, COG, Swiss-Prot, and Pfam databases annotated 50,398 (44.17%), 31,281 (27.42%), 53,366 (46.77%), 44,891 (39.34%), and 44,761 (39.23%) unigenes, respectively.

We then annotated the unigenes using the KEGG database to further elucidate their biosynthesis and metabolic pathways. The unigenes were assigned into five main categories representing 20 sub-clusters. The highest number of KEGG identifiers was in translation (3041 unigenes), followed by carbohydrate metabolism (2314 unigenes), folding, sorting, and degradation (2062 unigenes) (Figure 3A). To add further resolution to the findings, we also analyzed the sub-pathways of the significantly enriched metabolism pathways. For carbohydrate metabolism, the sub-pathway of amino sugar and nucleotide sugar metabolism (413) contained the most highly represented metabolites, followed by glycolysis and gluconeogenesis (401) and starch and sucrose metabolism (363). Among other secondary metabolites, the most prolific pathways were found to be involved in phenylpropanoid biosynthesis (243) and flavonoid biosynthesis (125). These findings indicate that carbohydrate metabolism, phenylpropanoid biosynthesis, and flavonoid biosynthesis play essential roles in the maturation of SYQ roots.

To illustrate the differences in the transcriptomic expression among the samples, we performed principal component analysis (PCA) and constructed a Venn diagram. PCA analysis showed that the three developmental stages were significantly separated in the confidence circle (Figure 3B). The gene expression patterns of the three developmental stages were different, and the sample selection was reasonable. The Venn diagram showed that there were 22,782 co-expressed genes in the three developmental stages and 21,167 specifically expressed genes in the T1 stage; this was significantly higher than the numbers of those in either the T2 or T3 stages (Figure 3C).

#### 2.3.2. Functional Analysis of DEGs

DEGs were determined based on per million reads (TPM) with |log2 FC| ≥ 1 and *p* < 0.05. In these three developmental stages, there were 16,164 DEGs identified between the T1 vs. T2 stages (1354 up-regulated and 14,810 down-regulated), 21,392 DEGs between the T1 vs. T3 stages (3279 up-regulated and 18,113 down-regulated), and 2595 DEGs between the T2 vs. T3 stages (1107 up-regulated and 1488 down-regulated). Hence, the number of DEGs in the T1 vs. T3 stages was higher than those in the other two combinations of stages, indicating that these two developmental stages had tremendous differential variation in their gene expression profiles (Figure 4A).

In order to confirm the relevance of the identified DEG-associated pathways in the formation of TRs, we employed KEGG enrichment analysis. The lowest score was found in the T2 vs. T3 comparison, indicating that many DEGs share the same expression patterns between these stages. The DEGs in the T1 vs. T2 stages were enriched in circadian rhythm-plant (map04712), flavonoid biosynthesis (map00941), and valine, leucine, and isoleucine degradation (map00280) (Figure 4B). DEGs in the T1 vs. T3 stages could not be significantly enriched to pathways (*p* < 0.5). However, the highest number of DEGs was found to be related to the spliceosome (map03040) (Figure 4C). DEGs in the T2 vs. T3 stages were significantly enriched in starch and sucrose metabolism (map00500), phenylpropanoid biosynthesis (map00940), plant hormone signal transduction (map04075), and zeatin biosynthesis (map00908) (Figure 4D). These results indicate that those genes related to flavonoid biosynthesis in the transition of the non-swelling stage turning to the initial swelling stage are significantly changed, while the biosynthesis of starch and sucrose, phenylpropanoid biosynthesis, and plant hormone signal transduction pathways are active in the initial swelling stage to the stable swelling stage. In the research results of other root crops, such as sweet potato and cassava, it was also found that starch, sucrose, and plant hormones were the core influencing factors of root development. They can form a regulatory network to affect development during the induction and swell of TRs. At the same time, flavonoid and phenylpropanoid biosynthesis are active throughout the growth cycle of SYQ. Therefore, we focus on genes involved in key pathways such as flavonoid and phenylpropanoid biosynthesis, starch and sucrose metabolism, and plant hormone signaling.

#### 2.3.3. Identification of TFs

In this study, all genes were predicted and classified into 34 transcription factors (TFs) families. The most abundant TF families were the *MYB* (168 genes), *bHLH* (108 genes), *AP2/ERF* (107 genes), *C2C2* (89 genes), *C2H2* (82 genes), *LBD* (73 genes), *bZIP* (66 genes), *C3H* (63 genes), *NAC* (62 genes), and *WRKY* (60 genes) families. MYB TFs transcription factors are widely present in plants and play essential roles in plant growth, development, and secondary metabolite generation. The *MYB* family of TFs consists of central regulators of phenylpropanoid derivatives metabolism. For example, MYB32 in *Arabidopsis* reduces the content of phenylpropanoid metabolites by decreasing the expression of essential structural genes of flavonoid and lignin synthesis pathways [14]. *MdMYB10* in apples promotes anthocyanin accumulation by increasing the expression of *DFR*, a necessary gene for anthocyanin synthesis in fruits [15]. *bHLH* TFs can act as transcriptional activators to activate the expression of other TFs in *Arabidopsis* to regulate hairy root development. The AP2/ERF family of TFs in trichomes may promote ripening and starch accumulation through the ABA signaling pathway [16]. The above suggests that these TFs also play an important regulatory role in the development and metabolite biosynthesis of TRs in SYQ. This requires further study.

#### 2.3.4. Weighted Gene Co-Expression Network Analysis (WGCNA)

To further investigate candidate unigenes related to starch and sucrose metabolism, flavonoid and phenylpropanoid biosynthesis, and plant hormone signal transduction pathways, WGCNA was performed to explore the co-expression gene network modules. As shown in Figure 5A, the co-expression network was constructed based on the 27,513 unigenes that remained after removing those with low expression from the total of 114,810 unigenes; those that remained were merged into 42 modules. Among the 42 modules, we selected the 24 modules with the greatest correlations with phenotype for subsequent analysis (Figure 5B). The correlation analysis showed that the steel blue module was significantly correlated with the content of starch. The turquoise module was clustered with the largest number of genes (3967) and was significantly correlated with the total flavonoids. The light cyan, midnight blue, and violet modules had significant positive correlations with the primary flavonoid components (catechin, kaempferol-3-O-rutinoside, and astragalin). Therefore, we conducted an in-depth study based on these five modules. The five modules contained flavonoid and phenylpropanoid biosynthesis-related DEGs (*CYP75A*, *COMT1*), starch and sucrose metabolism-related DEGs (*WAXY*, *glgA1*, *glgA2*, *glgA3*, *otsB3*, *TREH*, *TPS3*, *AMY2,* and *ISA2*), and plant hormone signal transduction-related DEGs (*IAA2*, *IAA6*, *SAUR8*) (Table 2). These gene IDs are listed in Appendix A. The results indicate that the DEGs in the modules made an important contribution to the traits observed at the T1, T2, and T3 stages, respectively. Subsequent analysis will also focus on the expression of these DEGs.

### 2.4. Metabolomic Analysis

#### 2.4.1. Identification of Metabolites

To better understand the metabolic changes that occur during the root development process, we profiled the metabolome analysis of the different samples via LC-MS. The results of partial least squares discriminant analysis (PLS-DA) showed that the metabolites were clearly separated among the three samples (Figure 6A). Hierarchical clustering of all metabolites showed that many were highly expressed at each development stage (Figure 6B). The blue and red bands represent low and high metabolite expression levels, respectively. From the distribution point of view, the composition of metabolites in stage T1 was significantly different from those of T2 and T3, while the similarity between T2 and T3 were higher.

A total of 577 metabolites were obtained from the three groups of samples. These metabolites were compared with the HMDB database to obtain the classification information of the metabolites (Figure 6C). The metabolite classifications primarily included lipids and lipid-like molecules (36.94%), phenylpropanoids and polyketides (14.86%), and organic acids and derivatives (14.23%). Among them, the flavonoids included 28 flavonoids and 6 isoflavones.

#### 2.4.2. Functional Annotation and Enrichment Analysis of Differentially Expressed Metabolites (DEMs)

The volcano plots show all the DEMs of T1 vs. T2, T1 vs. T3, and T2 vs. T3, including 174 DEMs (61 up-regulated, 113 down-regulated), 245 DEMs (86 up-regulated, 159 down-regulated), and 91 DEMs (32 up-regulated, 59 down-regulated), respectively (Figure 7).

KEGG enrichment analysis was performed based on the results of the differential metabolite annotation and the top 10 metabolites enriched pathways were selected for analysis. It was found that the enriched pathway results were consistent with the transcriptome results. KEGG enrichment analysis showed that the DEM enrichment of the T1 vs. T2 and T1 vs. T3 stages were most significant in ABC transporters (map02010), while the greatest enrichment of DEMs was found in arginine biosynthesis (map00220) (Figure 8A,B). The DEMs of the T2 vs. T3 stages were significantly enriched in the zeatin biosynthesis (map00908), starch and sucrose metabolism (map00500), plant hormone signal transduction (map04075), and phenylpropanoid biosynthesis pathways (map00940) (Figure 8C).

### 2.5. Combined Analysis of Transcriptome and Metabolome

#### 2.5.1. Expression Patterns of DEGs and DEMs Associated with Flavonoid and Phenylpropanoid Biosynthesis Pathways

To further explore the transcriptional regulators implicated in SYQ root development and quality, the major DEGs and DEMs that were enriched in flavonoid and phenylpropanoid biosynthesis, starch and sucrose metabolism, and plant hormone signal transduction pathways were analyzed. The results showed that 162 unigenes were predicted to be involved in the biosynthesis of flavonoids and phenylpropanoids. The differential expression analysis of samples was applied using DESeq2 software and 39 DEGs were screened (Figure 9). It was found that the expression of all selected DEGs involved in the flavonoid metabolism pathway was downregulated, including *CYP98A*, *FLS*, *CYP75A,* and *ANR* gene. In addition, the phenylpropanoid pathway also produced several phenolic compounds, such as lignin. Most of the genes were downregulated in the T2 and T3 stages, including *COMT*, *CYP81E*, and *PER* genes. It is reasonable to speculate that the number of lignified cells in TRs decreased in the T2 and T3 stages. There were nine DEMs significantly enriched in flavonoid and phenylpropanoid biosynthesis. Among them, caffeic acid is a precursor material of lignin biosynthesis, and its content decreased in T2 and T3 stages. The content of most flavonoid metabolites was decreased in the T2 and T3 stages, including (-)-catechin-3-O-gallate, 2-hydroxycinnamic acid, kaempferol 3-O-arabinoside, eriodictyol, and caffeic acid. The contents of the flavonoid metabolites 5,7,3′,4′,5′-pentahydroxyflavanone, glycitein-4′-O-glucuronide, and gossypetin-8-glucuronide-3-glucoside gradually increased during root development.

#### 2.5.2. Expression Patterns of DEGs and DEMs Associated with Starch and Sucrose Metabolism

The identified enrichment of DEGs related to starch and sucrose metabolism suggests their important role in root development. Here, a total of 145 genes in the transcriptome were annotated to starch and sucrose metabolic pathways, while 27 genes were found to be significantly differentially expressed between the three developmental stages (Figure 10). In sucrose biosynthesis pathways, the expression of the *SUS* gene was increased in the T2 and T3 stages. In starch biosynthesis pathways, the *WAXY*, *glgA1*, *glgA2*, *glgA3*, and *GBE1* genes were upregulated with root swelling. In starch catabolism pathways, the expression levels of *ISA1*, *ISA2*, *AMY2*, *BAM3*, *GUN12*, *GUN6*, and *GUN25* genes were upregulated. Overall, the number of upregulated genes was more significant than the number of downregulated genes. These data indicate that the upregulated genes facilitated the accumulation of starch and sucrose that promoted the swelling of TRs. Metabolic pathway analysis showed that the levels of levan, UDP-glucose, and amylose differed significantly during SYQ root development. The contents of these metabolites were increased in the T2 and T3 stages. These results were consistent with the transcriptome data. It is therefore inferred that root swelling in SYQ is directly associated with changes in carbohydrates.

#### 2.5.3. Expression Patterns of DEGs and DEMs Associated with Plant Hormone Signal Transduction Pathways

At the transcriptional level, a total of 162 unigenes were found to be possibly involved in seven plant hormone signal transduction pathways, including those coordinated by auxin (IAA), cytokinin (CTK), brassinosteroids (BR), jasmonic acid (JA), gibberellins (GA), abscisic acid (ABA), and salicylic acid (SA) (Figure 11). There were 22 DEGs related to IAA biosynthesis. In the IAA signaling pathway, 14 DEGs (including *IAA2*, *IAA3*, *IAA4*, *IAA6*, *IAA7*, *IAA10*, *SAUR 1*, *SAUR4*, *SAUR6*, *SAUR7*, *SAUR8*, and *ARF*) were upregulated in the T2 and T3 stages; this indicates that cell enlargement and plant growth were activated in T2 and T3 stages. At the same time, the expression levels of *GH3* and *AUX1* decreased. In the CTK signaling pathway, *B-ARR1*, *B-ARR2*, and *A-ARR1* were downregulated. In the GA and ABA signaling pathways, *GID1* and *ABF* were upregulated, which indicates that GA and ABA may promote root development. A total of 9 DEGs belonging to the BR signal transduction factors, including *BRI1*, *BSK1*, *BSK2*, *CYCD3*, and *TCH4*, were downregulated in the T2 and T3 stages. For the JA and SA signaling pathways, the *JAR1* and *TGA* genes were downregulated. At the metabolomic level, there was a significant difference in the contents of JA and cytokinins. Among the cytokinins, the content of adenine was upregulated in T2 and downregulated in T3. The content of JA was upregulated with prolonged root development.

#### 2.5.4. Connection Network between Gene Expression and Related Metabolite Accumulation

To clarify the relationships between the metabolites and genes involved in the pathways, we integrated the DEMs and DEGs to establish a Pearson correlation analysis (Figure 12). In the flavonoid and phenylpropanoid biosynthesis pathways, (-)-catechin-3-O-gallate, eriodictyol, and caffeic acid were highly correlated with the gene expression of *PER1*. In starch and sucrose metabolism pathways, levan and amylose were moderately correlated with both *otsB3* and *glgA1* expression. In the plant hormone signal transduction pathways, adenine and JA were correlated with *SAUR4* expression.

### 2.6. Verification of RNA-Seq Sequencing Data by qRT-PCR Analysis

To validate the expression profiles of the DEGs obtained by RNA-seq, 12 genes were selected for qPCR. The expression levels of the 12 genes were consistent with those of the RNA-seq data (Figure 13). This indicates that our sampling method used in this study and RNA-seq are both suitable for studying the transcriptomic patterns and developmental processes of SYQ TRs.

## 3. Discussion

Elucidating the molecular mechanisms associated with root swelling is important for improving the yield and quality of SYQ. Previous studies have focused on the growth, development, and metabolite biosynthesis of SYQ. Xiang et al. [17] studied the mechanisms underlying the development of TRs and the phenomenon whereby their contents of flavonoids and phenylpropanoids are affected by seasonal variation. However, seasonal variation may not fully reflect the changes across the whole growth period of SYQ. Additionally, Xiang et al. [18] used digital RNA-seq transcriptome profiling to reveal the developmental mechanism of calabash-shaped SYQ roots. However, there has so far been a lack of analyses conducted on the connections between key genes and metabolites. Therefore, it is necessary to investigate the regulatory networks and critical genes that control root swelling and metabolite biosynthesis in SYQ by integrating transcriptomic analysis and metabolomic analysis.

Through the anatomical observations made in the present study, we found that root swelling in SYQ was associated with vascular cambium activity. The number of lignified cells around the xylem bundles continued to decrease and the number of starch granules in parenchyma cells increased from the FRs to TRs; this is similar to results seen in cassava and sweet potato root development [19,20]. This suggests that the development of TRs may be related to lignin metabolism and starch biosynthesis. Next, we found that the T1 vs. T3 stages had the highest number of DEGs and DEMs. According to the KEGG enrichment analysis of the DEGs and DEMs, pathways related to flavonoid and phenylpropanoid biosynthesis, starch and sucrose metabolism, and plant hormone signal transduction were significantly enriched. A total of 105 DEGs and 13 DEMs related to these pathways were identified. Moreover, significant differences were found in these DEGs and DEMs between the different developmental stages. Accordingly, we combined the screened DEGs and DEMs across the three pathways to analyze the regulatory network of root development and metabolite biosynthesis.

### 3.1. Effect of Lignin and Flavonoid Biosynthesis on Root Swelling in SYQ

Of the number of lignin synthesis genes, *PER* is the largest. This suggests that there may be a certain internal relationship between lignin synthesis and the development of storage roots. Caffeic acid is a precursor metabolite of lignin synthesis, and its content was found to decrease with root development; this was consistent with the expression data of *PER*. These results reveal that a decrease in lignin was associated with the development of SYQ roots. In previous research on root swelling in other plants, the excessive expression of lignin has been found to decrease the contents of glucose and sucrose, which are not conducive to starch biosynthesis and cell expansion [21]. Storage root formation in sweet potatoes was also found to be accompanied by the downregulation of lignin biosynthesis and the upregulation of starch biosynthesis at the early stage [22]. Therefore, we speculated that lignin is needed to construct fibrous roots in the early stage, and that the lignin decreases as the contents of glucose and sucrose increase in the later stage.

WGCNA analysis further explored the DEGs related to flavonoid and phenylpropanoid biosynthesis (including *CYP75A* and *COMT1*). The expression of *CYP75A* was at a low level, and the expression of *COMT1* was upregulated. Through correlation network analysis, most of the metabolites were correlated with *PER1*. Bai et al. [23] previously investigated flavonoid metabolism in the roots and leaves of three-year-old SYQ. The results showed that *CHI* and *UFGT* were potential key genes involved in the biosynthesis of most flavonoids in SYQ. However, our transcriptome analysis did not find DEGs with *CHI* and *UFGT*, which suggests that there were no significant changes in *CHI* and *UFGT* across the three different developmental stages. A previous study revealed that flavonoids can act as IAA transport inhibitors to negatively regulate polar IAA transport and disturb the transport of endogenous auxins [24,25]. In the present work, flavonoid biosynthesis decreased while IAA content increased, which also confirmed the results. In general, most of the genes that regulate flavonoid biosynthesis were found to be downregulated. However, the contents of the total flavonoids in SYQ roots increased with prolonged root development. This may be due to the content of flavonoids being determined by two processes: biosynthesis and accumulation. In the first stage, the biosynthesis of flavonoid compounds in the root of SYQ was high. Although this gradually decreased in the middle and last stages, the gradual accumulation of metabolites still ultimately contributed to the observed increase in their contents.

### 3.2. Starch and Sucrose Metabolism Provides Materials for Root Swelling in SYQ

Starch and sucrose are the primary storage compounds in TRs, and play key roles in their growth [26,27]. The contents of starch, glucose, and sucrose provide sufficient energy and carbon skeletons used in tissue differentiation and cell division during early root development [28,29,30]. Some genes involved in starch and sucrose metabolism have been shown to be upregulated during the initiation and continued growth of TRs. UDP-glucose, which is produced from sucrose breakdown by *SUS*, is used for starch synthesis and promotes cell expansion [31]. In the present study, the content of UDP-glucose and the expression of *SUS* were found to be highest in the initial swelling stage, indicating that sucrose was hydrolyzed to synthesize starch. Similar results were obtained in a previous study, where *SUS* was found to be a key enzyme involved in the early development of radish storage roots [32]. WGCNA analysis found that the starch and sucrose metabolism-related DEGs were *WAXY*, *glgA1*, *glgA2*, *glgA3*, *otsB3*, *TREH*, *TPS3*, *AMY2*, and *ISA2*. The expression levels of those genes were found to be upregulated during the complete expansion phase to promote starch biosynthesis. The content of amylose, a precursor of starch synthesis, also increased. These results reveal that the biosynthesis of starch and sucrose increased continuously during the development of TRs. According to the correlation network, the amylose and levan metabolites were found to have a positive correlation with both *otsB3* and *glgA1*; this was consistent with WGCNA analysis. These results indicated that *otsB3* and *glgA1* may be key genes involved in regulating the development of TRs.

### 3.3. Plant Hormone Signal Transduction Regulates Root Swelling in SYQ

The formation and development of TRs results from the coordinated regulation of multiple endogenous hormones [33]. Our study showed that 40 DEGs were related to plant hormone signaling pathways. Among them, 22 DEGs were involved in IAA signaling, including *AUX1*, *IAA*, *ARF*, *GH3*, and *SAUR* genes, which were mostly upregulated during root swelling. These results suggest that IAA plays a key role in root development. WGCNA analysis found that plant hormone signal transduction-related DEGs were *IAA2*, *IAA6*, and *SAUR8*. Consistent with our findings, previous studies in sweet potatoes have shown that IAA levels were elevated during early TR formation [34]. In CTK signaling pathways, the *AHP* gene was upregulated, and the content of adenine was relatively higher in the T2 stage when compared to T1 and T3 stages. In addition, the expression levels of genes related to BR signaling (including *BRI1*, *BSK*, *CYCD3*, and *TCH4*) were significantly downregulated in the middle and last stages. The expression of the GA biosynthesis gene *GID1* was upregulated in the middle and last stages, while that of the ABA biosynthesis gene *ABF* was high in the last stage. Therefore, GA and ABA signaling may have potentially antagonistic interactions that modulate root swelling in SYQ. In other studies, the swelling of TRs has been found to be positively correlated with ABA in sweet potatoes and *Rehmannia glutinosa* [35]. In the current study, the content of JA increased during root development, indicating that JA may play a key role in root development. In the combined transcriptome and metabolome analysis, jasmonic acid and adenine had a correlation with *SAUR4* expression. In summation, the plant hormones IAA, GA, ABA, and JA appear to play important roles in the root development of SYQ.

### 3.4. Regulatory Process and Metabolite Biosynthesis of Root Swelling in SYQ

The regulation of root enlargement and the biosynthesis of metabolites involve lignin-related genes being downregulated in the TRs, thereby resulting in a decrease in lignin and the downregulation of metabolites related to flavonoids and phenylpropanoid biosynthesis. In addition, the expression levels of starch and sucrose genes were upregulated in TRs. In the initial swelling stage, sucrose is hydrolyzed into fructose to synthesize starch, which is conducive to cell division. The root development of SYQ is also related to IAA signaling. GA, ABA, and JA are closely associated with root development. In summary, these DEGs may play key roles in the root development of SYQ. Although the present study has uncovered the critical genes and metabolites expressed in tuberous roots, these results have been based on bioinformatic prediction and so lack adequate experimental evidence. Subsequently, we will further verify the function of these genes by biochemical, transgenic, and other methods, thereby providing new insights into the molecular mechanisms underpinning the regulation of tuberous root development.

## 4. Materials and Methods

### 4.1. Plant Materials

The SYQ roots used in this study were collected from Zhejiang Guangsheng Pharmaceutical Co., Ltd., Zhejiang, China (latitude: 28.8778, longitude: 118.7889; elevation: 77 m). The experimental materials were planted in April 2021 (T1), April 2020 (T2), and April 2019 (T3). They shared the same germplasm source, geographical location, and management practices. The SYQ roots were all harvested simultaneously in October 2021. According to the developmental status and age of the roots, they were allocated to non-swelling roots (T1), initially swelling roots (T2), and stable swelling roots (T3). Each stage had six biological replicates comprising 18 separate roots. These materials were divided into three parts: one part was rapidly frozen in liquid nitrogen and subsequently stored at −80 °C for transcriptome sequencing and metabolomic analysis; another part was used immediately for anatomical observations; and the last part was dried at 50 °C for 48 h to acquire a stable dry weight, which was then used for the analysis of flavonoid content.

### 4.2. Anatomical Observations

FAA fixative (formaldehyde:glacial acetic acid:ethanol = 5:5:90, *v*/*v*/*v*) was used to fix the cross sections obtained from the middle of the samples for 24 h. Paraffin wax was then used to embed these cross-sections. We cut 15-µm slices and placed the samples on silane-coated slides to fix them. Subsequently, the sections were dewaxed, dyed with 1% safranin O-fast green, decolorized, and then sealed. Finally, a Nikon eclipse 80i microscope and OPLENIC Pro software were used to observe the tissues. The numbers of ducts, starch, and thin-walled cells were counted in the fixation area.

### 4.3. Determination of Starch and Flavonoid Contents

A Starch Content Kit (Nanjing Jiancheng Bioengineering Institute, A148-1-1) was used to examine the total starch content. The soluble sugar and starch contents in the fresh sample were separated by 80% ethanol, after which the starch was further decomposed into glucose. The glucose was then quantified by anthrone colorimetry to calculate the content of related starch.

Roots were dried and then ground into powder to examine their content of flavonoids. The total flavonoid content was determined by ultraviolet spectrophotometry, as previously described [36]. Rutin was purchased from Push Bio-Technology (Chengdu, China) and used as the standard. Catechin, kaempferol-3-O-rutinoside, and astragalin were the primary flavonoid components of SYQ. They were determined by HPLC using the method proposed by Li et al. [37].

### 4.4. Transcriptome Sequencing

Total RNA was extracted from the T1, T2, and T3 samples using TRIzol^®^ Reagent (Plant RNA Purification Reagent for plant tissue) according to the manufacturer’s instructions (Invitrogen, Carlsbad, CA, USA). Genomic DNA was removed using DNase I (Takara). The integrity and purity of the total RNA were determined by 2100 Bioanalyser (Agilent Technologies, Inc., Santa Clara, CA, USA) and quantified using the NanoDrop 2000 (NanoDrop Thermo Scientific, Wilmington, DE, USA). High-quality RNA samples were used for library construction and sequencing.

The sequencing libraries were constructed with a TruSeqTM RNA Sample Preparation Kit (Illumina, San Diego, CA). Poly(A) mRNA was purified from the total RNA using oligo (dT) magnetic beads and then fragmented by fragmentation buffer. The mRNA was used as a template to synthesize double-stranded cDNA. Then the synthesized cDNA was subjected to end-repair, phosphorylation, and ‘A’ base addition according to Illumina’s s library construction protocol. After quantification, these samples were sequenced using the Illumina Hiseq NovaSeq 6000 sequencer (Illumina, San Diego, CA, USA) to generate a 150 bp paired-end sequence. RNA purification, reverse transcription, library construction, and sequencing were performed at Shanghai Majorbio Bio-pharm Biotechnology Co., Ltd. (Shanghai, China).

### 4.5. Transcriptome Data Analysis

Clean data obtained from T1, T2, and T3 stage roots were used to conduct de novo assembly with Trinity (http://trinityrnaseq.sourceforge.net/, accessed on 25 January 2022). After assembly completion, transcript sequence files in the FASTA format were obtained. The most extended transcript of each gene was extracted as a representative, called a unigene. The functions of assembled unigenes were annotated using the National Center for Biotechnology Information nonredundant protein sequence (NR), Protein family (Pfam), KOG/COG/eggNOG (Clusters of Orthologous Groups of proteins), Swiss-Prot (a manually annotated and reviewed protein sequence database), Kyoto Encyclopedia of Genes and Genomes (KEGG), and Gene Ontology (GO) databases. To identify the DEGs between two different samples, the expression level of each transcript was calculated according to the TPM method. RSEM (RNA-seq by Expectation-Maximization) was used to quantify gene abundances. Differential expression analysis was performed using the DESeq2 with *p*-value ≤ 0.05 and |log2FC| > 1. GO functional annotation and KEGG enrichment analysis was performed for the DEGs.

The WGCNA package was used for the co-expression network analysis. Here, *β* represents the soft threshold for the correlation matrix, which gives greater weight to the strongest correlations while maintaining the gene-gene connectivity. A *β* value of 16 was selected based on the scale-free topology criterion. The resulting adjacency matrix was converted to a topological overlap (TO) matrix via the TOM similarity algorithm. The genes were then hierarchically clustered based on TO similarity. The dynamic tree-cutting algorithm was used to cut the hierarchal clustering dendrogram. Modules were defined after decomposing/combining branches to reach a stable number of clusters. The minimum number of genes in the modules was set to 30. Modules with similar expression patterns were combined according to module eigenvalues (0.25).

### 4.6. Metabolic Analysis

Root material (50 mg) obtained from each sample was prepared and ground in liquid nitrogen. The metabolites were then extracted from them with 400 µL of methanol:water (4:1, *v*/*v*) solution. The mixture was allowed to settle at −20 °C before being treated by a Wonbio-96c high throughput tissue crusher (Shanghai Wanbo Biotechnology Co., Ltd., Shanghai, China) at 50 Hz for 6 min, followed by vortexing for 30 s, and ultrasound at 40 kHz for 30 min at 5°C. Then, the supernatant was carefully transferred to a sample vial for LC-MS/MS analysis, centrifuged at 13,000× *g* for 15 min, and finally analyzed.

The extracted root samples were separated on the UPLC system equipped with an ACQUITY BEH C18 column (130A, 100 mm × 2.1 mm i.d., 1.7 µm; Waters, Milford, MA, USA). The solvent gradient was changed as follows: from 0 to 3 min, phase A/phase B was 95:5 (*v*/*v*) to 80:20 (*v*/*v*); from 3 to 9 min, 80:20 (*v*/*v*) to 5:95 (*v*/*v*); from 9 to 13 min, 5:95 (*v*/*v*) to 5:95 (*v*/*v*); from 13 to 13.1 min, 5:95 (*v*/*v*) to 95:5 (*v*/*v*), from 13.1 to 16 min, 95:5 (*v*/*v*) to 95:5 (*v*/*v*) for equilibrating the systems. The sample injection volume was 2 µL and the flow rate was set to 0.4 mL/min. The column temperature was maintained at 40 °C. During the analysis, all samples were stored at 4 °C.

After UPLC-TOF/MS analysis, the raw data were imported into the Progenesis QI 2.3 (Nonlinear Dynamics, Waters, Milford, MA, USA) for peak detection and alignment. MS/MS fragmentation spectra and isotope ratio differences were acquired using the HMDB and Metlin databases. Metabolic changes between the developmental stages were compared using PLS-DA analysis. Metabolites with significant differences were screened with variable importance in projection (VIP) > 1 and an independent sample t-test (*p* < 0.05). Metabolite profiling and metabolomic data analyses were performed at Shanghai Majorbio Bio-pharm Biotechnology Co., Ltd. (Shanghai, China).

### 4.7. Integrated Analysis of Transcriptomic and Metabolomic Data

To further observe the changes and associations of the metabolites and genes, significant DEMs (VIP > 1, *p* < 0.05) and significant DEGs (|log2Fold Change| > 1, *p* < 0.05) in each of the three comparisons (T1 vs. T2, T1 vs. T3, T2 vs. T3) were used to construct a correlation network diagram with the Pearson Correlation Coefficient (|r| > 0.50). The correlation analysis and network plot were performed using Cytoscape (v 3.6.0) tools

### 4.8. Validation of Quantitative Real-Time PCR

To validate the results of RNA-seq, we conducted RT-qPCR assays using three biological replicates for each tissue sample and at least three technical replicates of each biological replicate. The total RNA was extracted from the collected plant materials using the OminiPlant RNA Kit (DNase I) (Cowin Bio., CW2598S). The cDNA was obtained by the FastKing RT Kit (With gDNase) (TIANGEN, KR116-02). SuperReal PreMix Plus (SYBR Green) (TIANGEN, FP205-02) was used for PCR amplification, in which the volume of each reaction was 20 µL, which contained 2 µL cDNA, 10 µL of 2× SuperReal PreMix Plus, 0.6 µL each of the forward and reverse primers (10 µmol/L), 2 µL of 50× ROX Reference Dye^△^, and 4.8 µL of ddH2O. The qPCR program was set to 95 °C for 15 min; 40 cycles of 95 °C for 10 s, and 60 °C for 10 s, followed by a melting curve analysis. Gene-specific primers were designed and synthesized in Shanghai Sangon Biotech Co. (Shanghai, China), Ltd., and listed in Appendix A. According to a previous study [38], each gene was normalized to the *GAPDH* internal control gene, and the fold change was calculated using the 2^−ΔΔCt^ method with three biological replicates.

### 4.9. Statistical Analysis

The statistical significance of the populations was calculated with one-way analysis of variance (ANOVA), and significance was indicated with different letters above the error bars. All data were expressed as the mean ± standard deviation (SD) of three repeated experiments (*n* = 3). An alpha value of *p* < 0.05 was considered statistically significant.

## 5. Conclusions

The development of SYQ TRs relies on a complex regulatory process. The lignin-related gene *PER1* was downregulated in TRs, resulting in a decrease in the root lignification of cells and a gradual expansion of the root. The decrease in lignin promotes the biosynthesis of starch, which provides energy for root swelling. The starch and sucrose biosynthesis genes otsB3 and glgA1 were upregulated, resulting in the increase of starch content, which is conducive to cell division. In addition, the plant hormones IAA, GA, ABA, and JA were found to be closely associated with root development. The IAA2, IAA6, SAUR4, and SAUR8 genes can accelerate root development by regulating hormone biosynthesis. Taken together, these results imply that the DEGs related to these pathways play an important role in the regulatory network of root swelling in SYQ. These findings not only potentially accelerate the process of the genetic improvement of SYQ, but also provide novel insights into the molecular regulatory mechanisms underlying TR morphogenesis in Chinese medicinal herbs.

## Figures and Tables

**Figure 1 molecules-28-02603-f001:**
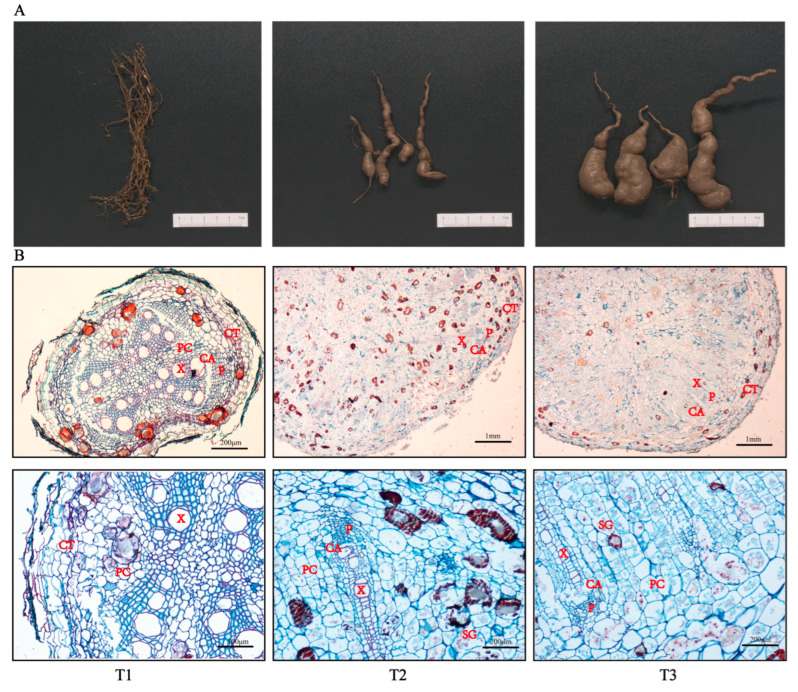
Physiological changes in different developmental stages of SYQ roots. (**A**) Morphology of roots. (**B**) Transverse sections of roots at different microscopic magnifications. Abbreviations: CA, cambium; CT, cortex; PC, parenchyma cell; X, xylem; P, phloem; SG, starch granules. The upward picture is the overall structure of SYQ roots at different developmental stages, and the picture below shows the details of the above image after zooming.

**Figure 2 molecules-28-02603-f002:**
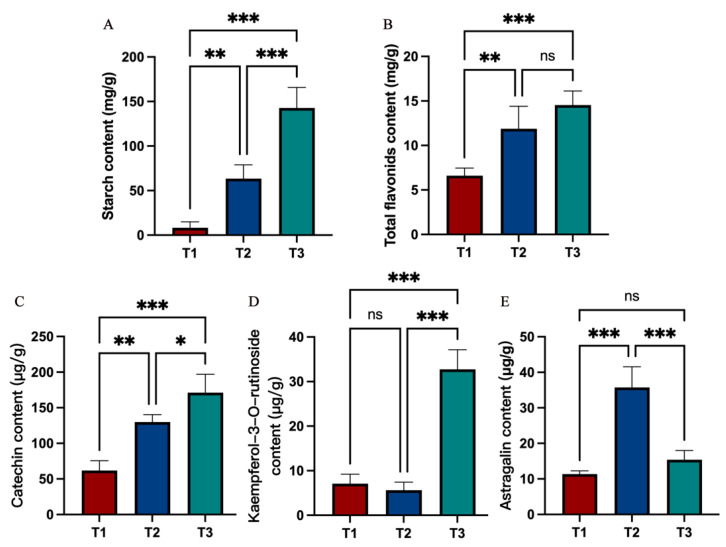
Variation in (**A**) starch, (**B**) total flavonoids, (**C**) catechin, (**D**) kaempferol-3-O-rutinoside, and (**E**) astragalin of SYQ roots at different developmental stages. Error bars indicate standard errors. The label (*) corresponds to significant differences (*p* < 0.05) by one-way analysis of variance (ANOVA), * *p* < 0.05, ** *p* < 0.01, *** *p* < 0.001, ns: not significant.

**Figure 3 molecules-28-02603-f003:**
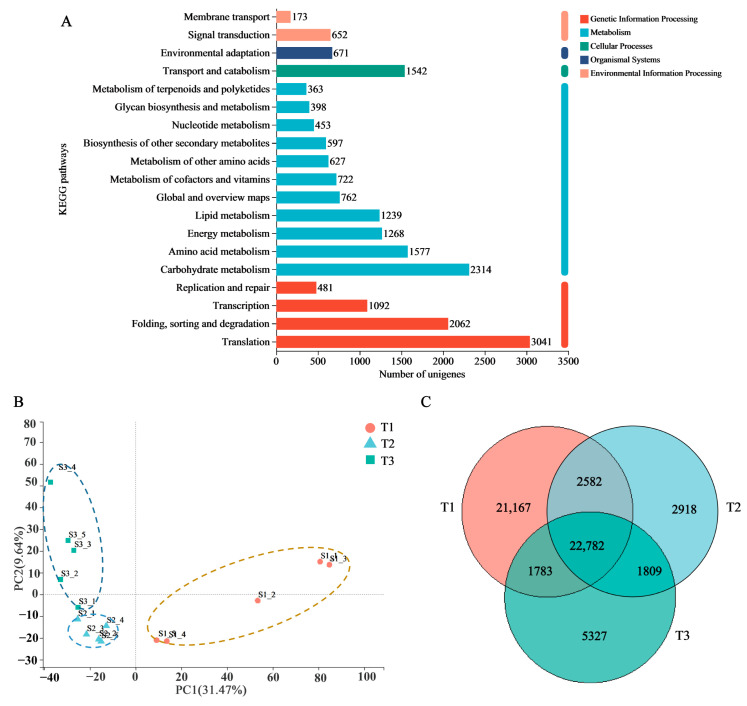
Expression analysis of unigenes in the roots of SYQ at different developmental stages. (**A**) KEGG annotation of unigenes. (**B**) PCA plot showing the clustering of transcriptomes. (**C**) Venn diagram of samples.

**Figure 4 molecules-28-02603-f004:**
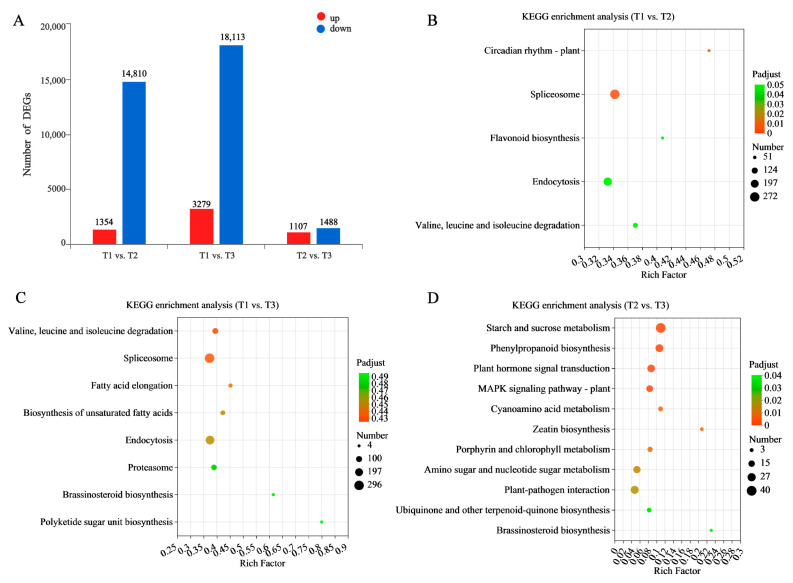
Significant DEGs analysis in three groups. (**A**) the number of DEGs. Red represents up-regulated DEGs, and blue represents down-regulated DEGs. KEGG analysis of the DEGs in (**B**) T1 vs. T2, (**C**) T1 vs. T3, (**D**) T2 vs. T3. The x-axis represents the richness factor, and the y-axis shows KEGG pathways. The color and size of the dots represent the *p*-value and the number of enriched DEGs, respectively. Rich factor means the ratio of the number of DEGs to the total number of genes enriched in a specific category.

**Figure 5 molecules-28-02603-f005:**
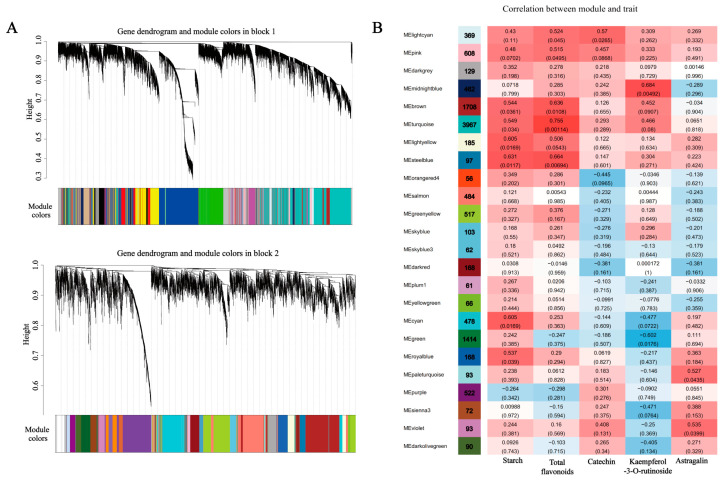
Co-expression network analysis for the construction of modules at three stages of SYQ. (**A**) Clustering dendrogram of SYQ root unigenes with assigned module colors. Genes clustering analysis based on TOM-based dissimilarity. Module divided by dynamic tree cut; different colors represented different modules. (**B**) Relationships between module eigengenes (ME) and traits at three stages. Each row corresponds to a module, and each column corresponds to a trait. Colors indicate the strength and direction of the correlation according to the color legend at right. The number in parentheses is the partial Pearson correlations and the corresponding *p*-values.

**Figure 6 molecules-28-02603-f006:**
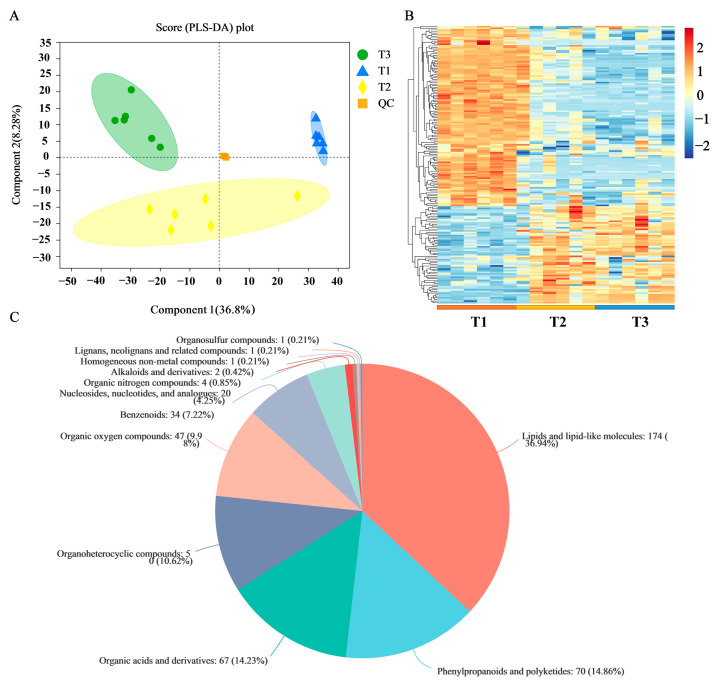
Metabolite annotation information of SYQ roots. (**A**) PLS-DA of metabolites identified in developmental stages. (**B**) Heatmap of all expressed metabolites at three developmental stages. The color indicates the level of relative content of each metabolite, from blue (low) to red (high). (**C**) Classification of overall metabolites.

**Figure 7 molecules-28-02603-f007:**
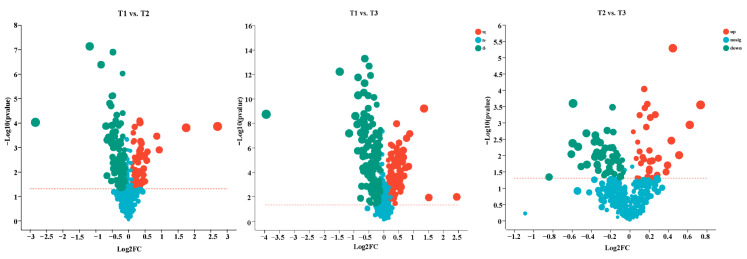
Volcano plots analysis of the DEMs in T1 vs. T2, T1 vs. T3, and T2 vs. T3, respectively. Red numbers and dots represent upregulated unigenes, blue numbers and dots represent downregulated unigenes, and gray dots represent unigenes with no change between comparing pairs.

**Figure 8 molecules-28-02603-f008:**
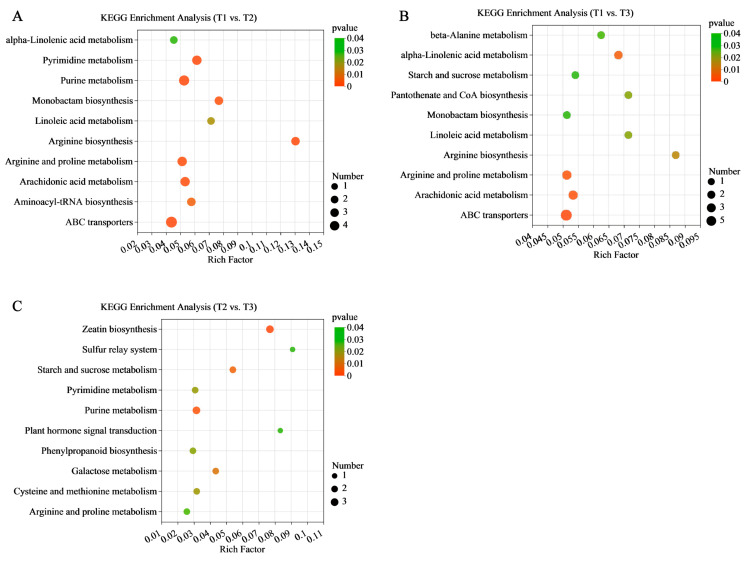
KEGG analysis of DEMs in (**A**) T1 vs. T2, (**B**) T1 vs. T3, (**C**) and T2 vs. T3. The *x*-axis represents the richness factor and the y-axis shows KEGG pathways. The color and size of the dots represent the *p*-value and the number of enriched differential metabolites, respectively. Rich factor means the ratio of the number of differential metabolites to the total number of metabolites enriched in a specific category.

**Figure 9 molecules-28-02603-f009:**
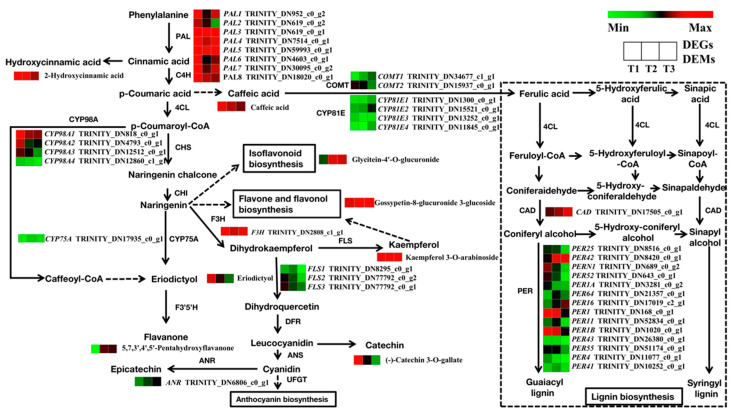
Regulatory network of flavonoid and phenylpropanoid biosynthesis in three different root samples. The color scale from green to red for the heatmap represents downregulated and upregulated expression. PAL, phenylalanine ammonia lyase; F3H, flavanone 3-hydroxylase; FLS, flavonol synthase; DFR, dihydroflavonol 4-reductase; ANR, anthocyanin in reductase; COMT, catechol-O-methyltransferase; CAD, cinnamyl-alcohol dehydrogenase.

**Figure 10 molecules-28-02603-f010:**
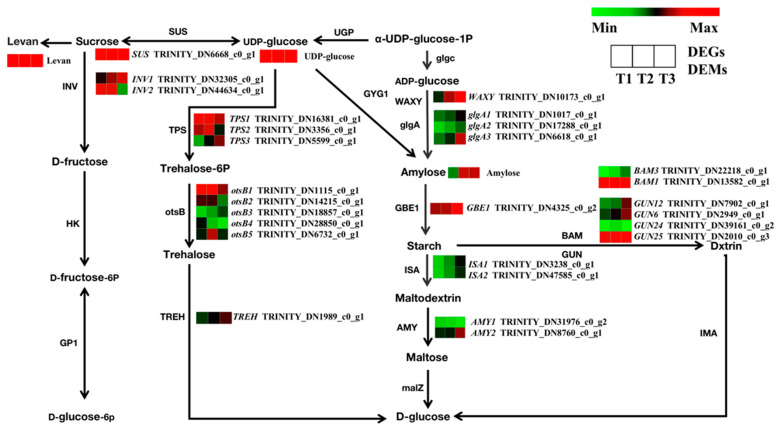
Regulatory network of starch and sucrose metabolism in three different root samples. The color scale from green to red for the heatmap represents downregulated and upregulated expression. SUS, sucrose synthase; INV, β-furan glycosidase; TPS, trehalose 6-phosphate synthase/phosphatase; TREH, α, α-trehalase; glgA, glycogen synthase; GBE1, 1,4-α-glucan branching enzyme; ISA, isoamylase; AMY, α-amylase.

**Figure 11 molecules-28-02603-f011:**
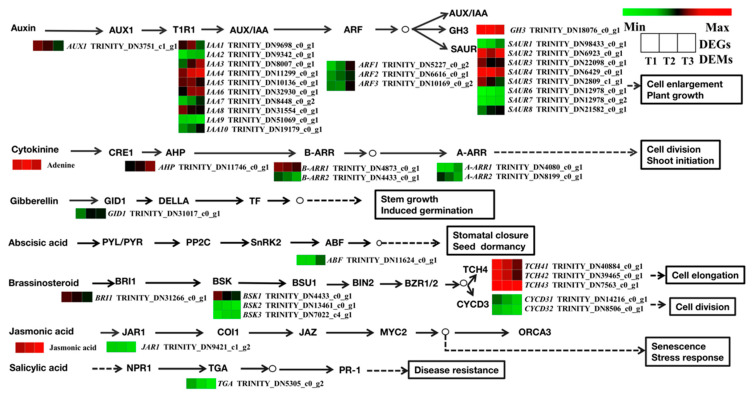
Regulatory network of plant hormone signal transduction in three different root samples. The color scale from green to red for the heatmap represents downregulated and upregulated expression. AUX1, auxin influx carrier; AUX/IAA, auxin-responsive protein/indole-3-acetic acid; ARF, auxin response factor; GH3, auxin-responsive GH3 gene family; SAUR, small auxin-up RNA family protein; A-ARR, two-component response regulator A-ARR family; B-ARR, two-component response regulator B-ARR family; GID1, gibberellin receptor GID1; ABF, ABA-responsive element binding factor; BRI1, protein brassinosteroid insensitive 1; TCH4, xyloglucan:xyloglucosyl transferase TCH4; CYCD3, cyclin D3; JAR1, jasmonic acid-amino synthetase; TGA, transcription factor TGA.

**Figure 12 molecules-28-02603-f012:**
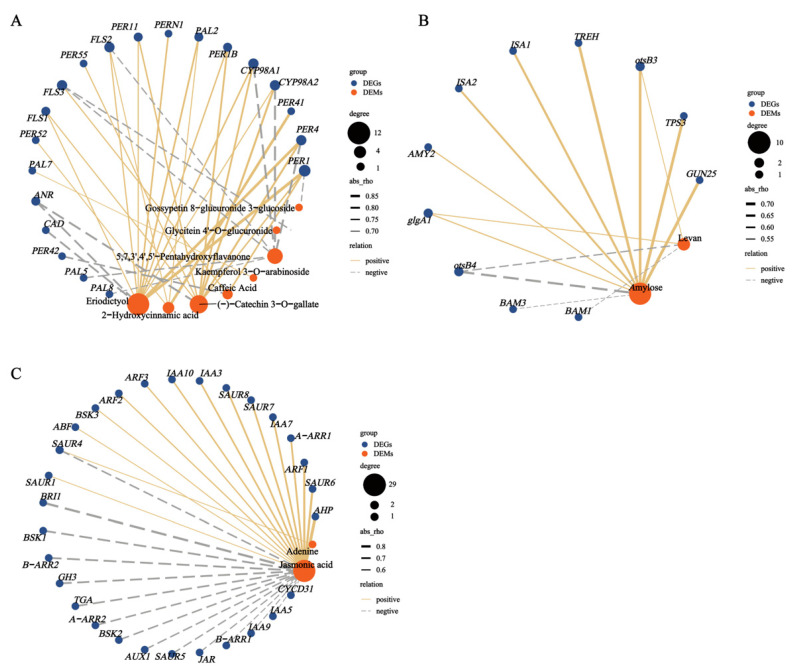
Correlation network between DEMs and screened DEGs of (**A**) flavonoid and phenylpropanoid biosynthesis. (**B**) starch and sucrose metabolism. (**C**) plant hormone signal transduction pathways. The blue circular nodes represent DEGs, the red circular nodes represent DEMs, the edges represent the correlation between nodes, and the width of each edge represents the magnitude of the correlation coefficient. The edges of the solid yellow line indicate a positive correlation between the gene and the compound, and the edges of the dashed gray line indicate a negative correlation between the gene and the metabolite.

**Figure 13 molecules-28-02603-f013:**
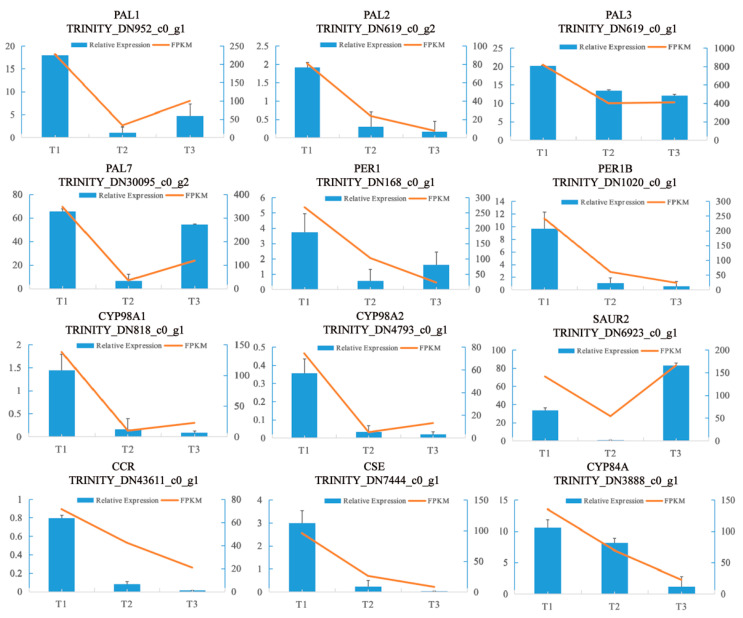
Verification of RNA-Seq sequencing data by the qRT-PCR assay. The left axis shows the relative expression, the right axis shows the fragments per kilobase million (FPKM) value, and the error line shows the SD value of the relative gene expression.

**Table 1 molecules-28-02603-t001:** Microscopic quantitative analysis of SYQ roots at different developmental stages.

	Vessel (/mm^2^)	Starch Granule (/mm^2^)	Parenchyma Cell (/mm^2^)
T1	208.33 ± 28.87 (a)	104.86 ± 50.94 (c)	3291.67 ± 187.64 (a)
T2	166.66 ± 14.43 (ab)	583.33 ± 118.15 (b)	358.33 ± 14.43 (b)
T3	116.66 ± 38.19 (b)	1016.67 ± 137.69 (a)	208.33 ± 28.87 (b)

Different letters correspond to significant differences (*p* < 0.05) by one-way analysis of variance (ANOVA).

**Table 2 molecules-28-02603-t002:** Distribution of DEGs related to (I) starch and sucrose metabolism, (II) flavonoid and phenylpropanoid biosynthesis, and (III) plant hormone signal transduction pathways in the five modules which were significantly correlated with the content of metabolite.

Module	I	II	III
Steel blue	0	0	0
Turquoise	9	0	1
Light cyan	0	1	2
Midnight blue	0	1	0
Violet	0	0	0

## Data Availability

The original contributions presented in the study are publicly available. This data can be found at NCBI repository BioProject ID: PRJNA880648 (https://www.ncbi.nlm.nih.gov/bioproject/PRJNA880648, accessed on 16 November 2022).

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
