# Peer review of "Integrative Analysis of the Transcriptome and Metabolome Reveals the Developmental Mechanisms and Metabolite Biosynthesis of the Tuberous Roots of Tetrastigma hemsleyanum"

_molecules, 2023, doi:10.3390/molecules28062603_

Round 1

Reviewer 1 Report

 Information available in the present form of the article looks very general and lack of innovation. It looks like a experiment report of metabolomic and transcriptomic results, with no definitive conclusive analysis and no confirmatory experiments. There are also flaws in the layout of the paper, much of previous researches analysis should be put into the discussion section. Many contents in the discussion section are simply repeated results, without in-depth analysis. This part needs to be improved critically. 

1. Table2 in line 99 should be changed to Table1. Some enzyme genes were not in italics in the manuscript.

2. Clearly, the authors blur the distinction between differentially expressed transcripts and differentially expressed genes. As shown in the heatmaps of metabolic pathways displayed in Figure 5 to Figure 7, there were many enzymes or functional proteins in metabolic pathways with multiple annotated transcripts with heterogeneity in their expression levels. The conclusions pointed by the authors regarding high and low variations in the expression levels of genes involved in metabolic pathways, which should be experimentally verified in conjunction with the expression level of each transcript itself.

3. The line 617-620, The DEGs were identified as related to root development in flavonoid and phenylpropanoid biosynthesis, starch and sucrose metabolism, and plant hormone signal transduction pathways among which PAL, CYP98A.... Candidate genes have been screened. So, I suggest further testing the function of genes in revealing the developmental mechanism and metabolite biosynthesis of tuberous roots.

4. As shown in Figure 8 and Figure 11, the authors ignored the absence of statistical significance in the enrichment analysis for the significantly enriched signaling pathways.

5.  The weighted gene co-expression network analysis described in the manuscript has significant defects. The algorithm for this analysis dictated that the expression profile used to construct the network should be complete because of the large number of hub genes in the core position of the network whose expression levels do not necessarily change appreciably, and improper filtering screening of these genes will disrupt intergenic connectivity and affect the reliability and robustness of gene expression network models. Second, the results of the correlation analysis between gene modules and traits did not exhibit the correlation between metabolism and gene modules. This makes this analysis appear redundant in the manuscript.

Reviewer 2 Report

Major issues:

1)     This MS had two sets of data. Although authors did correlation network analysis, data presented here actually are separated, not fully integrated. Therefore, the MS is indeed just a placement of the two data one by one, lacking convincing data to support the role of certain pathways in root development, because both data sets are generally too descriptive without any experimental validation. Suggest to focus on flavonoid and phenylpropanoid biosynthesis, starch and sucrose metabolism, and plant hormone signal transduction pathways, to draw a diagram harboring simultaneously both transcriptomic and metabolomic data.

2)     In M&M, authors did not provide clear information regarding sample preparation for LC-MS analysis, neither the reason why choosing roots at these three different developmental stages. Authors also did not provide detailed information regarding growth time period needed for T1, T2 and T3 roots. Are these three roots from the same year or season, or from different year and season? If it is the latter case, authors need also provide more information regarding harvest time and environment as well as other factors that could affect significantly the transcriptome and metabolome.

3)     In Introduction, authors did not give a brief introduction of the root development with anatomy features of T. hemsleyanum, which is important for readers to understand why authors chosen such three different developmental stages; In addition, a recent related reference was missing, which actually could be discussed in the Discussion section. Bai, Y.; Jiang, L.; Li, Z.; Liu, S.; Hu, X.; Gao, F. Flavonoid Metabolism in Tetrastigma hemsleyanum Diels et Gilg Based on Metabolome Analysis and Transcriptome Sequencing. Molecules 2023, 28, 83. https://doi.org/10.3390/molecules28010083.

4)     The English writing needs significant improvement. For example, line 468-469, what does this sentence mean? “Flavonoids have stability and the gene is down-regulated before the accumulation of metabolites. The amount of compound synthesis is reduced but still accumulates in the later process”.

Minor issues

1)     Fig. 1, lacking the labeling of each cell layers and tissues in the text; lacking the growth time for each sample. In addition, what is the differences between the upper and lower panel of B?

2)     Table 1, what do different letters in the parentheses mean?

Round 2

Reviewer 2 Report

Please redraw and revise the figure legends of Figures 9-12, make sure using the consistent color and format to show the up- and down- regulation of the DEGs and DEMs. Please pay attention to use italic capital for genes and non-italic capital for enzymes and proteins, to use red and green for up- and down- regulation of genes and metabolites. Therefore, red and green color either in word or box should not be carelessly used in these figures.
